# ARE LLMS EXPLOITABLE NEGOTIATORS ?

## ABSTRACT

Large Language Models (LLMs) have recently demonstrated surprising proficiency in strategic and social tasks, including bargaining and negotiation. Yet, their robustness in adversarial multi-agent interactions remains unclear. In this work, we study whether LLM-based agents are exploitable negotiators in the sense of game theory. To do so, we design a set of controlled game-theoretic environments — including auctions, markets, and public goods games — where Nash equilibria are analytically computable. These testbeds allow us to evaluate exploitability by comparing LLM outcomes against equilibrium predictions and against rational and adversarial opponents. Across multiple settings, we find that LLM negotiators systematically deviate from equilibrium: they tend to over-concede, are vulnerable to anchoring strategies, and often produce inefficient outcomes. Our results highlight that while LLMs can negotiate fluently, they remain strategically exploitable — raising concerns for their use in real-world interactions and opportunities for improving robustness through adversarial training and self-play.

## 1 INTRODUCTION

Negotiation is a fundamental component of multi-agent interaction. From resource allocation to market competition, agents must balance cooperation and competition, reasoning about others' incentives while simultaneously securing favorable outcomes. Success in negotiation requires not only generating persuasive offers and counteroffers, but also anticipating the strategies of others, guarding against manipulation, and ensuring that agreements remain stable over time.

Large Language Models (LLMs), trained on vast amounts of human text, appear capable of engaging in such tasks with remarkable fluency. They can produce offers, counteroffers, and persuasive arguments that resemble human negotiation behavior, and they often exhibit social reasoning patterns such as fairness, reciprocity, and compromise. These abilities suggest that LLMs could serve as powerful agents in multi-agent systems where communication and bargaining are central. However, beyond surface-level competence, a critical question remains:

**Are LLMs robust negotiators, or are they systematically exploitable?**

Answering this question requires going beyond linguistic plausibility and examining negotiation performance through a game-theoretic lens. In strategic interactions, robustness can be formalized as consistency with Nash equilibrium strategies and resistance to exploitation by adversarial opponents. For instance, in auctions, small deviations from equilibrium bidding strategies can lead to predictable financial losses; in public goods games, over-contributing can be systematically exploited by free-riders. Thus, assessing LLM negotiators requires quantifying not only what they say, but whether their strategies can withstand rational and adversarial play. In this work, we study LLM negotiators explicitly through the lens of exploitability. By situating them in canonical testbeds where equilibria are analytically tractable, we can precisely measure how far their behaviors deviate from rational play, and whether those deviations open the door to systematic exploitation.

Beyond evaluating whether LLMs "sound" like skilled negotiators, our approach reveals whether they actually act in ways that are strategically sound. Specifically, our contributions can be summarized as follows:

- Exploitability formalization: We operationalize the exploitability of LLMs as the deviation from Nash equilibrium utilities in resource allocation and market games.
- Controlled game-theoretic testbeds: We design experiments in auctions, Fisher markets, public goods games, market entry, and double auctions—each with computable equilibrium predictions.
- Systematic evaluation and findings: We compare LLM negotiators against rational baselines analyzing vulnerability to manipulation. Our results show that LLMs systematically deviate from equilibrium—e.g., overbidding in auctions, over-contributing in public goods games—and are thus exploitable in predictable ways.

The rest of the paper is organized as follows. Section 2 reviews related work on negotiation, exploitability, and LLM robustness. Section 3 introduces the necessary game-theoretic background and canonical examples. Section 4 details our experimental environments, agents, and evaluation metrics. Section 5 presents our results and analysis across bargaining, ultimatum, and coalition games. Finally, Sections 6 and 7 conclude with implications for robust negotiation and future directions.

## 2 RELATED WORKS

**Negotiation and Multi-Agent Interaction in AI** Negotiation and bargaining have long been studied within AI and multi-agent systems. Early approaches relied on symbolic reasoning and heuristic strategies for resource allocation and conflict resolution (Rosenschein & Zlotkin, 1994). With the advent of reinforcement learning, agents have been trained to negotiate and bargain through self-play and simulation (Foerster et al., 2018; Leibo et al., 2017). These works highlight the importance of equilibrium reasoning and adversarial robustness in multi-agent interaction. In contrast, our study shifts the focus from engineered negotiation agents to LLM-based negotiators, examining whether their human-like fluency translates into equilibrium robustness.

**Equilibrium Computation and Exploitability** Measuring robustness in multi-agent systems often involves analyzing exploitability. In imperfect-information games such as poker, exploitability has become the standard metric for evaluating strategy robustness, with milestones like Libratus and DeepStack achieving near-optimal play through equilibrium approximation (Brown & Sandholm, 2017; Moravcik et al., 2017). In cooperative-competitive games such as Hanabi, equilibrium-based evaluation has also been used to assess agent performance under uncertainty (Bard et al., 2020). Unlike prior work that focuses on RL-trained or search-based agents, we apply exploitability analysis to language-based negotiators, where strategies are implicit and not optimized for equilibrium.

**LLMs in Strategic and Social Tasks** Large Language Models have recently shown surprising ability to engage in social reasoning, persuasion, and negotiation. Meta's CICERO demonstrated that combining LLMs with planning and belief models can yield superhuman performance in Diplomacy (AI, 2022). Other work has explored LLM bargaining abilities in controlled negotiation tasks, showing that LLMs can produce linguistically coherent offers but may lack robustness against adversarial tactics (Pei et al., 2023; Wang et al., 2023). Our work differs by grounding the evaluation in canonical game-theoretic environments with known equilibria, enabling precise measurement of strategic exploitability.

**Robustness and Alignment of LLMs** Beyond negotiation, robustness has been a central concern in LLM alignment research. Studies have shown that LLMs are sensitive to adversarial prompts (Jones et al., 2023), framing effects and social manipulation (Park et al., 2023). Techniques such as adversarial training, self-play, and fine-tuning with equilibrium objectives have been proposed as potential mitigation strategies (Perez et al., 2022; Bai et al., 2022). In contrast to these general robustness studies, our work specifically quantifies

exploitability in adversarial negotiation settings, linking alignment concerns directly to game-theoretic vulnerabilities.

## 3 BACKGROUND

### 3.1 STRATEGIC GAMES

We model negotiation settings as finite normal-form games. A game is defined as a tuple

$$G = (N, \{A_i\}_{i \in N}, \{u_i\}_{i \in N}),$$

where:

- $N = \{1, \ldots, n\}$ is the set of players,
- $A_i$ is the action set of player $i$, with joint action space $A = \times_{i \in N} A_i$,
- $u_i : A \to \mathbb{R}$ is the utility function for player $i$.

A (mixed) strategy for player $i$ is a probability distribution $\sigma_i \in \Delta(A_i)$. A *strategy profile* is $\sigma = (\sigma_1, \ldots, \sigma_n)$.

A *Nash equilibrium (NE)* is a strategy profile $\sigma^\star$ such that

$$u_i(\sigma_i^\star, \sigma_{-i}^\star) \geq u_i(\sigma_i, \sigma_{-i}^\star), \quad \forall i \in N, \ \forall \sigma_i \in \Delta(A_i),$$

where $\sigma_{-i}$ denotes the strategies of all players except $i$. In equilibrium, no player can unilaterally improve their payoff.

### 3.2 EXPLOITABILITY

Following the tradition in multi-agent reinforcement learning and imperfect-information games, we define *exploitability* as the gap between an agent's achieved utility and the best-response utility against it.

Given a candidate strategy profile $\sigma$, the best response of player $i$ is

$$BR_i(\sigma_{-i}) = \arg \max_{\sigma_i \in \Delta(A_i)} u_i(\sigma_i, \sigma_{-i}).$$

The exploitability of $\sigma$ for player $i$ is then

$$\text{Exploit}_i(\sigma) = u_i(BR_i(\sigma_{-i}), \sigma_{-i}) - u_i(\sigma_i, \sigma_{-i}).$$

At equilibrium, $\text{Exploit}_i(\sigma^\star) = 0$ for all $i$. Positive values indicate vulnerability. In our setting, we use this to measure how exploitable LLM-generated strategies are when faced with rational or adversarial opponents.

### 3.3 CANONICAL GAME EXAMPLES

**Auctions.** In a first-price sealed-bid auction with private valuation $v$, the symmetric equilibrium bidding strategy is

$$b(v) = \frac{n-1}{n} v.$$

Deviating from this strategy (e.g., by bidding too high) yields predictable losses.

**Public Goods Game.** Each player contributes $c_i \in [0, E]$ from endowment $E$. The total contribution is multiplied by $\alpha < n$ and redistributed equally:

$$u_i(c_1, \ldots, c_n) = E - c_i + \frac{\alpha}{n} \sum_{j=1}^{n} c_j.$$

The dominant-strategy equilibrium is $c_i = 0$ for all $i$ (free-riding).

**Alternating-Offer Bargaining.** Two players divide a resource of size 1 over alternating rounds. With discount factor $\delta$, the equilibrium predicts delayed concessions, with the first proposer receiving approximately

$$\frac{1 - \delta}{1 - \delta^2}.$$

**Ultimatum Game.** A proposer offers a split $(x, 1-x)$. The responder accepts if $x \geq \epsilon$ and rejects otherwise. Subgame-perfect equilibrium prescribes minimal offers just above the acceptance threshold.

**Coalition Games.** In transferable-utility coalition games, the *core* is the set of allocations where no coalition can improve upon the proposed split. LLM-based negotiators may fail to respect such stability notions, leading to exclusionary dynamics.

### 3.4 NEGOTIATION IN AI AND LLMS

Traditional AI approaches often use reinforcement learning and self-play to train agents toward equilibrium strategies, with notable successes in poker, Go, and Diplomacy. By contrast, LLMs are pretrained on human text and not explicitly optimized for payoff-maximization. Their behavior reflects human priors such as fairness norms, cooperation, and persuasion tactics, which may improve social plausibility but also lead to systematic deviations from equilibrium play and hence exploitable vulnerabilities.

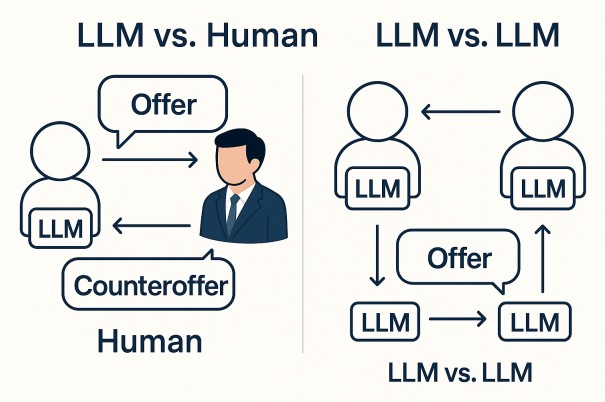

Figure 1: Illustration of Various Negotation Settings

# 4 EXPERIMENTAL SETUP

## 4.1 GAME ENVIRONMENTS

We evaluate LLM negotiators in three canonical game-theoretic environments where Nash equilibria are analytically tractable:

- **Alternating-Offer Bargaining.** Two agents divide a fixed endowment of 10 tokens over multiple rounds. In equilibrium, the first proposer secures a slight advantage (53–47 under our discounting scheme). We measure whether LLMs reproduce this equilibrium split or deviate toward alternative focal points.

- **Ultimatum Game.** A proposer offers a division of 10 tokens, which the responder accepts or rejects. The subgame-perfect equilibrium predicts that proposers take nearly all tokens, as responders accept any nonzero allocation. This game tests whether LLMs adopt equilibrium-maximizing proposals or deviate toward fairness.

- **Coalition Formation.** Three agents divide 15 tokens, where any coalition of two can finalize an allocation, excluding the third. Equilibria correspond to allocations in the *core*, though exclusionary coalitions are possible. This setting tests robustness in multi-agent negotiations and vulnerability to exploitation.

These environments were chosen because they represent canonical bargaining, ultimatum, and coalition dynamics, and because equilibrium predictions are well-defined, enabling precise measurement of exploitability.

## 4.2 AGENTS

We compare the following agents:

- **LLM negotiators:** LLaMA-7B, Mistral-7B, and Falcon-7B, prompted with natural language instructions to propose offers, respond to offers, and form coalitions.

- **Rational agent:** A scripted baseline that plays the equilibrium strategy when available (e.g., proposer keeps all in ultimatum).

- **Exploitative agent:** A scripted anchor designed to maximize payoff against over-conceding LLMs in coalition settings.

- **No-Regret bidder (NR3):** For auctions, a simple algorithmic baseline that bids proportionally to valuation ($b(v) = v/2$), converging to equilibrium.

We evaluate both *self-play* (same model against itself) and *cross-play* (different models against each other or scripted baselines), as well as heterogeneous coalition games.

## 4.3 METRICS

We measure outcomes along three axes:

- **Equilibrium deviation.** Difference between observed payoffs and theoretical Nash equilibrium predictions (measured in tokens).

- **Exploitability gap.** Payoff loss relative to a rational best response, capturing how much utility LLMs forgo when faced with adversarial or rational opponents.

- **Outcome properties.**

–  *Efficiency:* Total welfare relative to the maximum achievable.
–  *Fairness:* Distributional balance, measured via the Gini coefficient.
–  *Coalition stability:* Frequency of exclusionary coalitions and payoff disparities across members.

### 4.4  IMPLEMENTATION DETAILS

Each interaction round is conducted through structured natural language prompts (e.g., `"Propose a split of 10 tokens"`). Model responses are parsed into structured actions (offers, acceptances, coalition choices) using deterministic templates. Invalid responses are re-prompted until a valid action is produced. All experiments are repeated across 100 episodes to ensure robustness.

**Reflective Prompting.**  To test whether LLMs can internalize equilibrium reasoning across repeated interactions, we implement a *ReflectiveLLMAgent* wrapper. This agent prepends an evolving "reflection memo" to the prompt at the beginning of each block of episodes. The memo summarizes past performance (average distance from equilibrium bids, win rate, profit, and over/underbidding frequency) and provides heuristic guidance on how to adapt. A fixed header also reminds the agent of the equilibrium bidding rule:

> "In an auction with $n$ bidders, the equilibrium bid is $(n-1)/n \cdot v$. Reflect on past mistakes and adjust toward profit-maximizing bids. Do not bid 0 unless really necessary."

This design allows the agent to iteratively update its behavior across repeated episodes, simulating a form of meta-cognitive learning through natural language reflection.

The prompt structure therefore consists of three components:

1. **Reflection Memo:** dynamically updated after each block, summarizing recent errors and profits.
2. **Guidance Header:** static reminders about equilibrium play and profit maximization.
3. **Context:** the current game state (e.g., valuation in an auction, outstanding offers in bargaining).

If the LLM produces invalid outputs (e.g., non-numeric bids), the wrapper automatically normalizes the action into a valid response (clipping to $[0, v]$ in auctions). This ensures comparability across models while still allowing us to observe systematic deviations from equilibrium.

## 5  RESULTS AND ANALYSIS

### 5.1  NEGOTIATION GAMES

In alternating-offer bargaining, illustrated in Figure 2, we observed two distinct regimes. In self-play, LLMs converged to symmetric 50–50 splits, consistent with fairness norms but deviating from the Rubinstein equilibrium of 53–47. However, in cross-play matchups, initiating models frequently secured the entire resource. For example, LLaMA-7B consistently captured all tokens against both Mistral-7B and Falcon-7B, leaving the opponent with zero.

When symmetric outcomes occurred, the deviation from equilibrium was stable: one agent under-received by 3 tokens while the other over-received by 3. In asymmetric cross-play, deviations were even larger, with one agent capturing 100% of tokens despite equilibrium predicting nontrivial concessions.

These results suggest that while LLMs can converge to fairness norms in symmetric self-play, cross-play introduces hierarchical patterns where stronger or initiating models dominate weaker ones. This indicates fragility in strategic robustness: equilibria are not consistently reproduced, and bargaining outcomes are shaped by linguistic anchoring rather than rational play.

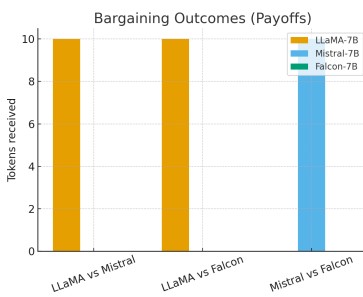

Figure 2: Bargaining game outcomes in cross-play matchups. Bars indicate final payoffs (in tokens) for each agent. Initiating models (e.g., LLaMA-7B) frequently secured the full allocation, while opponents received none.

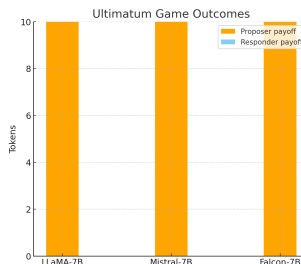

Figure 3: Ultimatum game outcomes. Proposers (LLaMA-7B, Mistral-7B, Falcon-7B) consistently kept the full endowment, while rational responders received zero.

## 5.2 ULTIMATUM GAMES

As shown in Figure 3, across all tested models, proposers consistently secured the full endowment (10 tokens), while rational responders received zero. These outcomes match the subgame-perfect equilibrium prediction of the ultimatum game.

Unlike bargaining, ultimatum games induced equilibrium-like behavior. However, this consistency arises not from nuanced strategic reasoning but from the linguistic simplicity of the task ("take all" offers are easy to express and accept). Thus, robustness in this case reflects structural simplicity rather than generalized equilibrium reasoning.

## 5.3 COALITION GAMES

Coalition formation experiments, illustrated in Figure 4, produced heterogeneous outcomes. In LLM-only coalitions, allocations were uneven but not maximally exploitative. In mixed settings with rational or exploitative agents, however, LLMs were frequently disadvantaged, with stronger or adversarial players capturing disproportionate payoffs.

These results reveal that LLMs are vulnerable to exclusionary strategies. Notably, even rational agents—expected to secure stability—were disadvantaged in some runs, indicating that LLM negotiation heuristics can destabilize coalition outcomes.

## 5.4 CROSS-GAME COMPARISON

Taken together, the results reveal a spectrum of LLM negotiation behavior:

- In bargaining (Figure 2), LLMs sometimes coordinate on fairness norms but often fall into asymmetric, exploitable splits.

- In ultimatum games (Figure 3), LLMs reproduce equilibrium-like outcomes, though likely due to structural simplicity rather than deeper reasoning.

- In coalition games (Figure 4), LLMs display vulnerability to exploitation, with stronger or adversarial agents consistently capturing disproportionate payoffs.

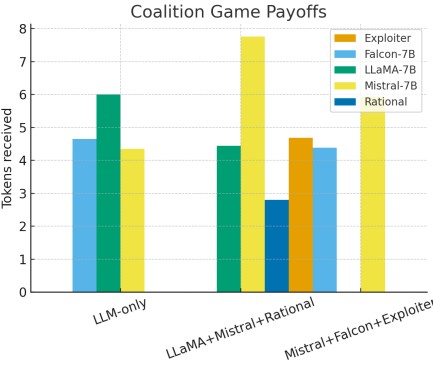

Figure 4: Coalition game outcomes across three settings: LLM-only, mixed with a rational agent, and mixed with an explicit exploiter. Stronger or adversarial agents consistently extracted disproportionate payoffs, leaving others disadvantaged.

Overall, LLMs can sustain socially plausible or equilibrium-like outcomes in simple settings, but they lack robustness in adversarial or multi-agent coalition contexts. These findings highlight systematic exploitability and emergent hierarchies across model matchups.

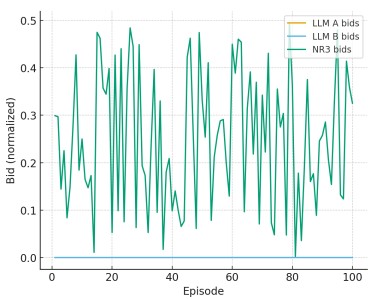

Figure 5: Bidding evolution of the No-Regret bidder across rounds.

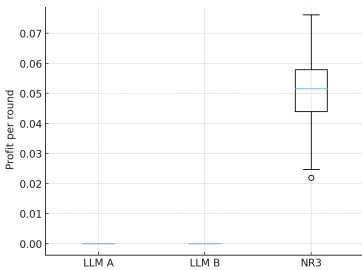

Figure 6: Profit distribution for each agent in a multi-LLM game.

Figures 5 and 6 further illustrate convergence dynamics and profit distributions, complementing the negotiation results discussed above.

## 6 DISCUSSION

### 6.1 WHY ARE LLMS EXPLOITABLE?

Our results suggest that LLM negotiators are systematically exploitable for two key reasons.

First, LLMs inherit a *training bias toward cooperation and agreement*. Pretraining on human text encourages models to favor polite, cooperative, and consensus-oriented responses. While this improves fluency and persuasiveness in natural dialogue, it biases agents toward over-concession in bargaining and over-contribution in public goods settings. Such biases produce outcomes that are socially plausible but strategically suboptimal.

Second, LLMs lack *recursive reasoning about opponents*. Strategic robustness requires anticipating how others will respond and best-responding in turn. Human game-theoretic reasoning often involves at least first- or second-order beliefs ("I think that you think that I will..."). LLMs, however, appear to operate myopically: they respond to the immediate prompt without deeply modeling adversarial counter-strategies. This leads to vulnerabilities against anchoring, exclusionary coalitions, and exploitative scripted agents.

### 6.2 IMPLICATIONS FOR ALIGNMENT

The exploitability of LLM negotiators raises broader questions for AI alignment. On the one hand, *exploitable agents* may be safer than *manipulative agents*: if LLMs systematically give up resources, they pose less risk of extracting unfair advantage from humans. On the other hand, exploitable systems can be co-opted in high-stakes domains—for example, market interactions, diplomatic negotiations, or resource allocation in multi-agent systems. In such settings, an exploitable LLM could incur predictable losses for its user, destabilize agreements, or entrench power asymmetries when deployed alongside more strategic agents.

More broadly, our findings underscore a gap between *linguistic competence* and *strategic competence*. LLMs can speak the language of negotiation but lack the robustness guarantees expected in adversarial or competitive domains. Bridging this gap is essential if LLMs are to serve as trustworthy decision-making partners.

### 6.3 MITIGATION STRATEGIES

Several avenues may help mitigate exploitability:

- **Adversarial training.** Exposing LLMs to systematically exploitative strategies during fine-tuning can help them learn to resist anchoring, over-concession, and exclusionary coalitions.
- **Self-play and population training.** Training LLMs through repeated self-play against diverse opponents may foster equilibria that are more robust than fairness heuristics. This approach has been effective in reinforcement learning for games such as Go and Poker.
- **Equilibrium-aware prompting.** Incorporating equilibrium strategies into prompts (e.g., reminders of Nash predictions or rational best responses) may nudge LLMs closer to theoretically grounded play, as we observed in reflective prompting for auctions.
- **Hybrid architectures.** Combining LLMs with algorithmic modules (e.g., regret minimizers or equilibrium solvers) could yield systems that retain fluency while ensuring game-theoretic soundness.

These directions highlight that improving robustness is not merely a matter of scaling model size, but of integrating adversarial, game-theoretic, and equilibrium-aware signals into training and deployment.

## 7 CONCLUSION

Our study shows that LLMs, while linguistically competent, are not robust negotiators. Across bargaining, ultimatum, and coalition games, they systematically deviate from equilibrium predictions and can be exploited by rational or adversarial opponents. In bargaining, models either collapse to fairness norms (50–50) or allow initiators to capture all resources. In ultimatum games, they reproduce equilibrium-like outcomes but for trivial structural reasons. In coalitions, they are vulnerable to exclusionary strategies, leading to systematic power asymmetries. These findings highlight the need for robustness evaluation in multi-agent LLM systems. Negotiation is not only a test of linguistic fluency but also of strategic soundness. Without equilibrium-aware reasoning, LLMs risk being systematically exploited in domains ranging from economic markets to diplomacy. Future work should therefore pursue adversarial fine-tuning, self-play training, and hybrid equilibrium-aware architectures to bridge the gap between fluent language use and robust strategic reasoning.

**LLM Usage.**
We acknowledge the use a large language model as assistant for writing, search and coding.

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
