# OpenReview forum: "Are LLMs Exploitable Negotiators ?"
_ICLR.cc/2026/Conference — ICLR 2026 Conference Withdrawn Submission_

### Official Review · Reviewer_mRZU · 2025-10-27

**Soundness:** 1
**Presentation:** 1
**Contribution:** 1
**Rating:** 0
**Confidence:** 5

**Summary:**

The paper asks whether LLM are 'exploitable negotiators' and proposes evaluating models in canonical analytically tractable games. It defines exploitability through best-response gaps, then report results in bargaining, ultimatum, and 3-player coalition forming. They claim LLM deviate from equilibrium.

**Strengths:**

The question itself is clear and there's a good motivation to use and connect with game-theory. There's a sensible choice of canonical games to study.

**Weaknesses:**

Overall, there's a few major concerns on this paper, which makes me feel like the paper, in its current form, is not ready for publication.

- Scope mismatch and overclaims: the abstract and contributions promise auctions, fisher markets, public goods, market entry, and double auctions. However, nearly all are NOT present in the experimental analysis. In particular, Section 4 evaluates only bargaining, ultimatum, and coalitions; auctions appear tangentially in Figure 5-6 without much exposition beyond the two lines given, and all other settings are not examined. This is a significant overclaim of contributions and raise questions on how the introduction paragraphs are written. This also fundamentally undermines the central, cross-domain claim.

- Lack of novelty and exposition of prior work: there has been significant literature on LLMs bargaining and various setting. In particular, it is studied by Abdelnabi et al. (2023), Bianchi et al. (2024) (which provides exact study on ultimatum and negotiation), Deng et al. (2024), etc. I omit of a full list and citations of papers on LLMs and negotiation that are readily available through a simple keyword search. The authors include none of these papers in their related works section, which highly overlaps with what they are doing. Without such comparison and exposition it is not apparent what this paper achieves beyond the existing works.

- The key point in the paper from my reading is the exploitability definition in sec 3.2, where authors define exploitability as best-response utility gap. However, in the results section, this was never actually plotted or tabulated. The figures show raw payoffs or equilibrium deviations but there's no mention or plot of this central notion the paper define.

- The experimental results seem poorly presented. Figures are descriptive; there are no CIs, hypothesis tests, sample-size justifications. Models itself is less specified - the paper consistently talks about "LLAMA 7b". It is unclear what version of the llama model the authors are referring to. If the authors are referring to llama 1 in 2023, this is a legacy model that carries much less weight given how many updates it underwent. Along the lines, details like prompt templates are not found.

- Design decisions are unclear in the absence of details. e.g. the authors claim that "Invalid responses are re-prompted until a valid action is produced." With the example prompt having words like (line 252) "Do not bid 0 unless really necessary". These doesn't seem like a natural part of the game and could change the game itself, which needs a more detailed discussion and justification.

- Game theoretic correctness and lack of specifications. The authors mention rubinstein model, but its formulation in line 151 lacks a concrete $\delta$. Similarly the $\epsilon$ is not specified in the ultimatum analysis in line 154. These precise game parameters being omitted cast doubt on internal validity. In addition, in the intro the authors state the exploitability as "we operationalize exploitability as deviation from Nash equilibrium utilities", while in section 3.2 they defined it as a best response gap. This seems inconsistent.

**Questions:**

It will be helpful if the authors can directly respond to my points in the weakness section above. In particular, how this work is novel and what aspect and discovery is not examined in past work; detail model and setup specifications; better statistics and reporting of results, and how they tie together with proposed concepts like exploitability; more details on the auction setup beyond line 363-364.

---

### Official Review · Reviewer_Zb9j · 2025-10-31

**Soundness:** 3
**Presentation:** 1
**Contribution:** 1
**Rating:** 2
**Confidence:** 4

**Summary:**

The paper studies how LLM agents can be exploited in negotiation. The paper experiments with a set of controlled game-theoretic environments, including auctions, markets, and public goods games, where Nash equilibria are analytically computable. The experiments compare LLM outcomes against equilibrium predictions and against rational and adversarial opponents. The paper finds that LLM negotiators systematically deviate from equilibrium.

**Strengths:**

- The games provide numerical ground truths for evaluation.

**Weaknesses:**

- The writing of the paper needs to be improved to a large extent. Examples: Figure 1 is never referenced in the text. It is also not related to the setup of the paper, which does not involve human-to-LLM negotiation or a group of LLMs. Figures 5 and 6 are not well explained in the text. The experiments, metrics, and terms could be written in a more accessible way.

- The related work of the paper is missing. Previous work [1] studied how greedy agents can exploit others during negotiation. Given the large number of papers already on LLM and negotiation (e.g., https://arxiv.org/pdf/2402.05863), the novelty of this paper is not clear, as it relies on canonical games and limited evaluation.

- The paper experiments with small models (7B). These models may have limitations in arithmetic skills, which could explain why they are vulnerable to exploitation. The results may not transfer to advanced models. Findings such as ("Unlike bargaining, ultimatum games induced equilibrium-like behavior. However, this consistency arises not from nuanced strategic reasoning but from the linguistic simplicity of the task (“take all” offers are easy to express and accept)") are probably very specific to these models.

- The paper does not give detailed insights into the reasons for failures and supporting examples/experiments to the discussion.

- The discussion of the paper ("Equilibrium-aware prompting") is not generic in practice with more complex real-world scenarios. In addition, findings such as ("LLMs can speak the language of negotiation but lack the robustness guarantees expected in adversarial or competitive domains") might not hold for reasoning and advanced models.

- The paper is more specific to a scripted Exploitative agent and does not include a setup where agents play against an exploitative LLM agent. The latter can be interesting to observe how LLMs devise exploitation strategies.

[1] https://proceedings.neurips.cc/paper_files/paper/2024/hash/984dd3db213db2d1454a163b65b84d08-Abstract-Datasets_and_Benchmarks_Track.html

**Questions:**

- Are there any empirical observations from models' textual output to back up the conclusion that models converge to fairness norms?

---

### Official Review · Reviewer_ZpVu · 2025-11-01

**Soundness:** 1
**Presentation:** 1
**Contribution:** 1
**Rating:** 0
**Confidence:** 4

**Summary:**

The paper evaluates whether LLMs are strategically exploitable in negotiation settings by comparing their behavior against Nash equilibrium predictions in game-theoretic environments (alternating-offer bargaining, ultimatum games, coalition formation). Results show LLMs deviate from equilibrium, converging to fairness norms (50-50 splits) or allowing initiators to capture all resources in bargaining, and being vulnerable to exclusionary strategies in coalitions.

**Strengths:**

1. This challenge is important for deployment in multi-agent systems.
2. Organisation of the paper is decent, the concepts explain adequately.
3. It looks at both self-play and cross-play in a systematic comparison.

**Weaknesses:**

1. No references in either the introduction or discussion. Claims like "LLMs have demonstrated surprising proficiency in strategic tasks" (line 10) and "training bias toward cooperation" (line 372) lack supporting citations.
2. Some of the references seem made-up. A) "Alex Jones et al. Adversarial prompting for language models. arXiv preprint arXiv:2302.09912, 2023.” This arXiv paper does not exist and I see no researcher with this name in the field. B) "Jack Pei et al. Llms as negotiators: Language models in bargaining games. In NeurIPS Workshop on Cooperative AI, 2023.” There was no workshop with that name at NeurIPS 2023. Paper does not seem to exist either. C) "Y. Wang et al. Testing strategic coordination of llms in multi-agent settings. In ICLR Workshop on LLM Agents, 2023.” Does not seem to exist either.
4. Section 2 is superficial and missing key recent work on:
   - LLM strategic reasoning and game-playing (e.g., Brookins & DeBacker 2023 on GPT cooperation, Horton 2023 on economic games, Kolbeinsson & Kolbeinsson 2024 Adversarial Negotiation Dynamics in Language Models)
   - Multi-agent LLM systems (AgentVerse, MetaGPT, etc.)
   - Game theory in AI alignment literature
   - Human behavioral game theory (Camerer 2003, Kahneman work on anchoring)
5. Limited experiments with only 3 games tested, 100 episodes each. Seems to be single-run. Results shown without error bars, confidence intervals, or significance tests. Impossible to assess whether observed differences are meaningful. Method details claims "100 episodes to ensure robustness" (line 244) but provides no variance estimates.
6. Figure 1 is not very informative and is not referenced or discussed anywhere in the main text.
7. Figures 2 and 3 have extremely small text and the legend hides the result of one of the bars. A bar plot it not an informative way to show these results as half the bars are 0 and do not appear at all.
8. Figure 5 seems to be very noisy while most of the series are either below the quantisation scale or are 0. It is not very informative.
9. Abstract and contributions claim evaluation in "auctions, Fisher markets, public goods games, market entry, and double auctions" (line 53), but experiments only cover bargaining, ultimatum, and coalition games. Section 3.3 defines these other games but never evaluates them.
10. No comparison to human behavior in these games (extensive experimental economics literature exists). Are LLMs more or less exploitable than humans? No comparison to RL-trained agents or other AI negotiators beyond trivial scripted baselines.

**Questions:**

1. Can you show variance across episodes and models?
2. What are the exact prompts used for each game?
3. Can you provide statistical significance tests and confidence intervals?
4. How does LLM exploitability compare to human subjects in these same games?
5. Why no comparison to other LLM negotiators (CICERO, existing work)?
6. What is the discount factor δ for alternating bargaining?
7. Did reflective prompting actually improve performance in any setting?

---

### Official Review · Reviewer_yhGM · 2025-11-01

**Soundness:** 1
**Presentation:** 1
**Contribution:** 1
**Rating:** 0
**Confidence:** 4

**Summary:**

The paper examines whether large language models behave as robust or exploitable negotiators in simple game-theoretic settings.
The authors test LLaMA-7B, Mistral-7B, Falcon-7B in bargaining, ultimatum, and coalition-formation games.
The results show that the models tend to over-concede and can be exploited by scripted agents or stronger models.
The paper argues that this stems from biases towards cooperation and a lack of recursive opponent modelling.

**Strengths:**

1. The study addresses an interesting and timely question about strategic robustness in LLMs, linking negotiation behaviour to formal game-theoretic analysis.
2. The use of canonical games with known Nash equilibria provides a clean and interpretable evaluation framework.

**Weaknesses:**

1. Several references appear questionable or unverifiable. For example, *Wang et al., 2023* (“Testing strategic coordination of LLMs in multi-agent settings”) could not be found, suggesting that parts of the bibliography may be hallucinated.
2. Figure 1 is unclear, and the caption does not explain what is being illustrated. It is difficult to understand how the figure relates to the experiments.
3. The models used are not properly specified. While LLaMA-7B, Mistral-7B, and Falcon-7B are mentioned, the paper omits version details and decoding parameters, which prevents reproducibility.
4. Figure 2 presents results as bar charts, which is not ideal for multiple match outcomes. A matrix of pairwise results would communicate asymmetries more clearly.
5. The number of episodes (100) is small, and no confidence intervals or variance estimates are reported. Without statistical detail, the robustness of the results is uncertain.
6. The paper promises a wide range of environments (auctions, markets, and public-goods games) but reports results only for bargaining, ultimatum and coalition settings. This inconsistency reduces the scope of the contribution.
7. Quantitative reporting of exploitability is limited. The paper defines the metric formally but does not provide actual numerical exploitability values per game or per model. This makes it hard to judge how large or meaningful the deviations are.
8. The discussion of limitations is brief and superficial. The paper would benefit from explicitly addressing model scale, prompt design biases and the narrow range of tested games.

**Questions:**

1. Are all references verified and publicly available?
2. Are all models run against all others, including self-play and both initiator roles?

---

### Note · Authors · 2025-11-12

I have read and agree with the venue's withdrawal policy on behalf of myself and my co-authors.